# CNT-molecule-CNT (1D-0D-1D) van der Waals integration ferroelectric memory with 1-nm² junction area

Thanh Luan Phan [1,5], Sohyeon Seo [2,5], Yunhee Cho[2,3], Quoc An Vu[3,4], Young Hee Lee [3,4], Dinh Loc Duong [3,4] ✉, Hyoyoung Lee [2,3] ✉ & Woo Jong Yu [1] ✉

The device's integration of molecular electronics is limited regarding the large-scale fabrication of gap electrodes on a molecular scale. The van der Waals integration (vdWI) of a vertically aligned molecular layer (0D) with 2D or 3D electrodes indicates the possibility of device's integration; however, the active junction area of 0D-2D and 0D-3D vdWIs remains at a microscale size. Here, we introduce the robust fabrication of a vertical 1D-0D-1D vdWI device with the ultra-small junction area of 1 nm² achieved by cross-stacking top carbon nanotubes (CNTs) on molecularly assembled bottom CNTs. 1D-0D-1D vdWI memories are demonstrated through ferroelectric switching of azobenzene molecules owing to the cis-trans transformation combined with the permanent dipole moment of the end-tail -$CF_3$ group. In this work, our 1D-0D-1D vdWI memory exhibits a retention performance above 2000 s, over 300 cycles with an on/off ratio of approximately $10^5$ and record current density $(3.4 \times 10^8 \text{ A/cm}^2)$, which is 100 times higher than previous study through the smallest junction area achieved in a vdWI. The simple stacking of aligned CNTs $(4 \times 4)$ allows integration of memory arrays (16 junctions) with high device operational yield (100%), offering integration guidelines for future molecular electronics.

Molecular electronics have been considered as a fundamental building block for miniaturizing electronic devices by using sub-nanometer scaled active components[1–6], which have high potential for practical applications such as field-effect-transistors (FETs), sensors, detectors, and memory devices[7–10]. To date, various sophisticated techniques have been utilized to form molecular devices such as electromigrated nanogaps[11–13], scanning probe microscopy (SPM)[14,15], mechanically-controllable break junctions[16], and e-beam lithography to form molecular-scale gaps by precise cutting of carbon nanotubes (CNTs) or graphene electrodes[5,17]. However, manufacturing electrodes with gaps on a molecular scale is a formidable task, which limits the high-density integration of molecular electronic devices.

Meanwhile, a van der Waals heterostructures (vdWHs) of 2D materials have been proposed for the realization of various devices such as vertical tunneling FETs[18,19], photo-detectors[20–26] diodes[27–31], and memory devices[32–37]. Recently, vdW stacking has been expanded to the van der Waals integration (vdWI) of various dimensional materials such as nanowire-graphene (1D-2D) field-effect-transistors (FETs) for ultra-high speed transistor applications[38], nanowire-oxides (1D-3D) with very high stability[39], graphene-oxide (2D-3D) memories

[1]Department of Electrical and Computer Engineering, Sungkyunkwan University, Suwon 16419, Republic of Korea. [2]Department of Chemistry, Sungkyunkwan University, Suwon 16419, Republic of Korea. [3]Center for Integrated Nanostructure Physics (CINAP), Institute for Basic Science (IBS), Suwon 16419, Republic of Korea. [4]Department of Energy Science, Sungkyunkwan University, Suwon 16419, Republic of Korea. [5]These authors contributed equally: Thanh Luan Phan, Sohyeon Seo. ✉e-mail: ddloc@skku.edu; hyoyoung@skku.edu; micco21@skku.edu

with a high on/off ratio exhibiting excellent durability[40], carbon nanotube-TMD (1D-2D) transistors with 1 nm gate lengths for excellent switching characteristics with a near-ideal subthreshold swing of 65 mV/dec[41], and quantum dot-graphene (0D-2D) photodetectors with ultrahigh gain[42]. Molecular electronics have also achieved vdWI by stacking a vertically aligned molecular layer between bottom and top electrodes (BE and TE, respectively). A current is driven from the BE to the TE through the molecule layer, exhibiting various electronic and optoelectronic properties[43–45]. For example, photo-switchable flexible memories and high-yield, ambient-stable molecular devices have been demonstrated using graphene-molecule-graphene (2D-0D)[43] and metal–molecule–metal (3D-0D)[44] heterostructures, respectively. However, the active vdWI junction areas of 2D-0D and 3D-0D are limited to microscale sizes, posing a formidable challenge for future high-density device integration.

A 0D-1D vdWI is expected to provide advantages regarding integration of ultra-small-scale devices and their corresponding electrical characteristics through nanoscale vdW junctions, but this has not yet been demonstrated. In this work, we demonstrate a 1D-0D-1D vdWI memory fabricated by vdW stacking of top carbon nanotubes (CNT$_T$) above a layer of self-assembled molecules (SAM) covering bottom CNTs (CNT$_B$). Our CNT$_T$-SAM-CNT$_B$ 1D-0D-1D vdWI forms the smallest reported junction area of 1 nm$^2$ at the cross point of the top and bottom CNTs. The ferroelectric polarity transition of the azobenzene molecules between the trans-isomer (high resistance state-HRS) and cis-isomer (low resistance state-LRS) states provides further insight using experimental and theoretical simulations. Using this platform for building multi-cross-junction arrays (up to 4 × 4 CNTs, 16 active domains), our vertical 1D-0D-1D vdWI molecular device provides a clear path toward the assembly of ultra-short junction areas for future molecular electronics applications.

## Results

### CNT$_B$-M/CNT$_T$ vdWI device fabrication and characterization

Figure 1a shows a schematic illustration of the fabrication steps for the CNT$_B$-M/CNT$_T$ (0D-1D-0D) vdWI device. The individually aligned metallic-CNT$_B$ (m-CNT$_B$) was transferred from a CVD-grown sample onto a 300 nm-thick SiO$_2$/Si substrate using a conventional wet transfer technique (Fig. 1a (i)). The CNT$_B$ is covalently functionalized by the azobenzene molecules that allow ultra-thin mono-molecular self-assembly[43] (Fig. 1a (ii), details are provided in the Methods section and Supplementary Figs. 1–3). Then, CNT$_T$ is stacked on CNT$_B$-M by the well-known dry transfer method, forming a vdW contact (Fig. 1a (iii)). Figure 1b shows the Raman mapping images at each fabrication step. The G-peak intensity (-1590 cm$^{-1}$) of the CNT was measured under an excitation laser wavelength of 532 nm. The diameter of the CNT was

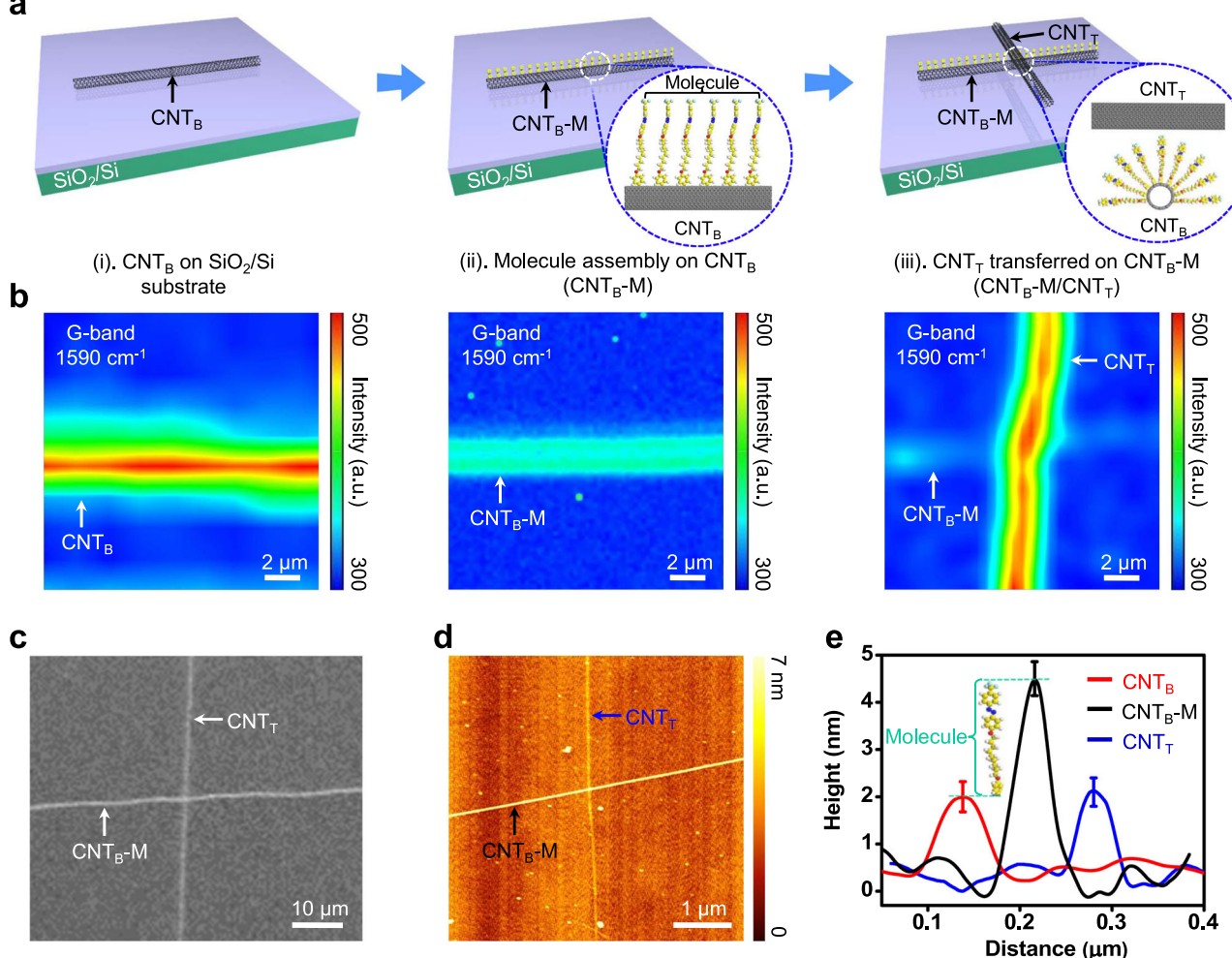

**Fig. 1 | Device fabrication process and characterizations. a** Schematics of the fabrication steps for the CNT$_B$-M/CNT$_T$ vdWI device (1 × 1 CNT array). The inset image in (ii) and (iii) shows the schematics of the molecule assembly on CNT$_B$. **b** The Raman mapping corresponding to each step in (**a**), where the G-band (1590 cm$^{-1}$) peaks are observed at a laser wavelength of 532 nm. **c** The SEM images for the CNT$_B$-M/CNT$_T$ vdWI device in (**a**, **b** (iii)). **d** AFM image of the CNT$_B$-M/CNT$_T$ single cross-junction corresponding to (**c**). **e** Height profile distributions of CNT$_B$ (red curve), CNT$_B$-M (black curve) and CNT$_T$ (blue curve) are plotted from the AFM function in (**d**).

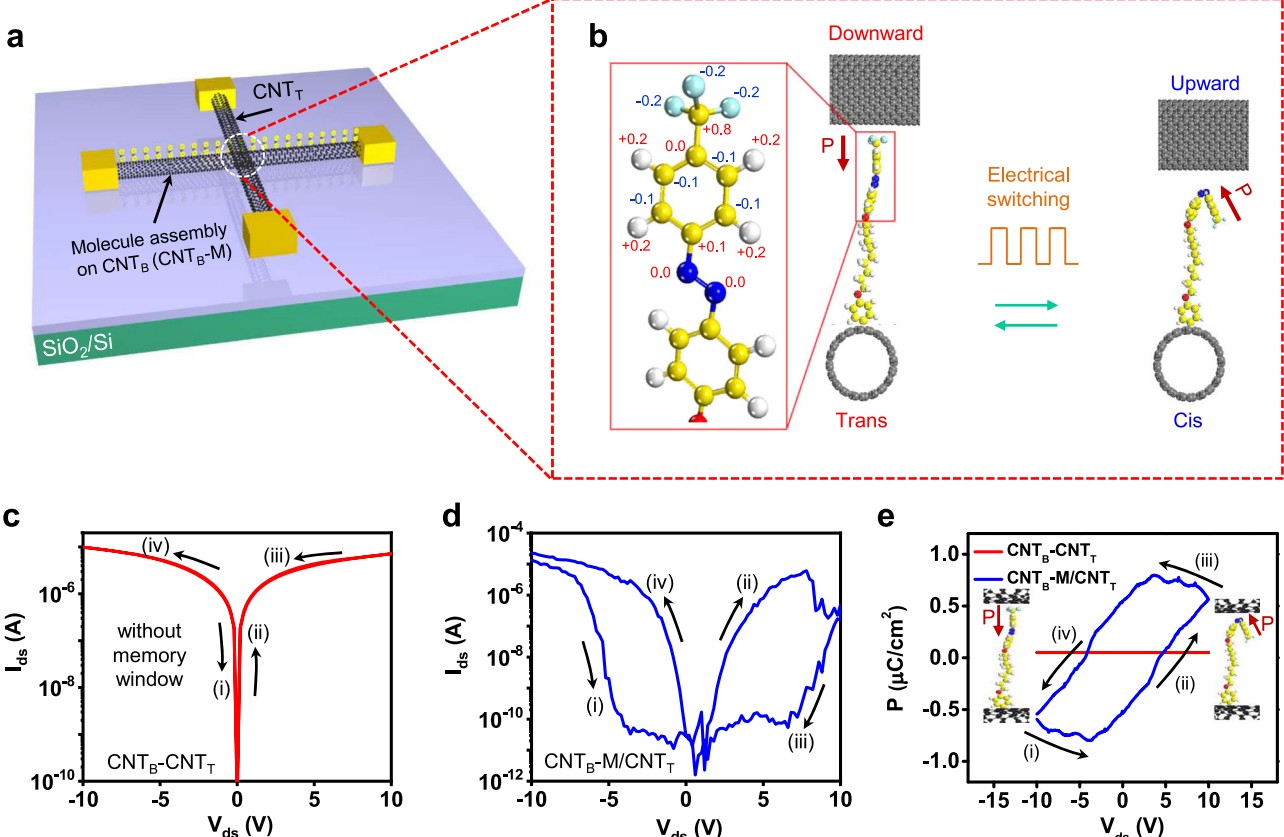

**Fig. 2 | Device structure and ferroelectric characteristics of the CNT$_B$-M/CNT$_T$ vdWI. a** Three-dimensional view of the CNT$_B$-M/CNT$_T$ vdWI. The layers (from bottom to top) are CNT$_B$, molecule (M), and CNT$_T$. **b** DFT simulation model of the molecule polarization change based on trans and cis isomers in CNT$_B$-M/CNT$_T$ vdWI according to electrical switching. **c** Output characteristic ($I_{ds}$-$V_{ds}$) curves of CNT$_B$-CNT$_T$ device (without molecule assembly). Forward (i, ii) and reverse (iii, iv) scans indicating no memory windows at V$_{ds}$ sweep ranges = ±10 V (negligible interface charge states at the interlayer interface of the vdWI). **d** Output characteristic ($I_{ds}$-$V_{ds}$) curves of CNT$_B$-M/CNT$_T$ vdWI device (with molecule assembly). The switching of molecules between trans and cis states leads to the generation of a memory window. **e** Ferroelectric polarization versus bias for CNT$_B$-CNT$_T$ (red curve) and CNT$_B$-M/CNT$_T$ (blue curve). The measurements were conducted on the same device as in (**c, d**).

approximately 1 nm based on the radial breathing mode (RBM) peak estimation (Supplementary Fig. 4)[46], indicating that the critical dimensions of the CNT electrodes and active area at the CNT-molecule-CNT junction are ~1 nm and ~1 nm², respectively. Covalent C–C sp³ bonds formed between carbon atoms C=C sp² of CNT$_B$ and aryl radicals (azobenzene molecular) were produced through a chemical reaction, yielding CNT$_B$-M, whose electrical characteristics were slightly lower than the original CNT$_B$ (Supplementary Fig. 5). Therefore, the Raman G-band intensity of CNT$_B$-M was reduced after the chemical reaction, as shown in Fig. 1b (i-iii). This is in contrast to the original CNT$_B$ and CNT$_T$, which showed similar intensities because there is no aryl radical chemical reaction[43]. The effects of azobenzene molecules chemisorbed by CNT$_B$ were further characterized by Raman and Fourier transform infrared (FTIR) spectroscopy (Supplementary Fig. 4, 6). Figure 1c shows a scanning electron microscopy (SEM) image of a single-cross junction of the CNT$_B$-M/CNT$_T$ (1D/0D) vdWI device at an accelerating voltage of 1 kV, corresponding to Figs. 1a and 1b (iii). An atomic force microscopy (AFM) image was taken, and a height profile distribution analysis was performed to confirm the topology and thickness of each component in our sample (CNT$_B$, azobenzene molecule, and CNT$_T$ (Fig. 1d, e)). Here, the thicknesses of pure CNT$_B$, CNT$_B$-M (after molecule assembly), and pure CNT$_T$ were measured as ~1.6 nm (red), ~4.2 nm (black), and ~1.6 nm (blue), respectively, indicating that the length of the azobenzene molecule is ~2.6 nm (Fig. 1e), which is in good agreement with the height of azobenzene molecule in DFT calculation (2.8 nm). It is noted that the molecules surrounding

the CNT$_B$ (inset of Fig. 1a-iii)[47] allow the top molecule to stand vertically by the support of side molecules. The thickness (diameter) of CNT calculated from the AFM and Raman spectroscopy is slightly different, which is attributed to the roughness of the SiO$_2$ interface.

## Theoretical simulation and ferroelectric characterization

Figure 2a shows a three-dimensional image of our CNT$_B$-M/CNT$_T$ ('-' = chemical contact, '/' = vdW physical contact) vdWI device structure. The azobenzene molecule used in our experiments contain a −CF$_3$ group at the end of the tail (Fig. 2b, left panel). This group exhibits a strong dipole characteristic with negative (positive) charge located at the F (C) atoms along the axis of the azobenzene, as shown by the density functional theory (DFT) calculations in the Methods section. It is known that the azobenzene molecules transformed between cis and trans states by external bias[48], inducing the ferroelectric switching memory behavior in CNT$_B$-M/CNT$_T$ vdWI. The thickness change of CNT$_B$-M/CNT$_T$ vdWI of trans–cis transition of azobenzene was measured to ~0.6 nm (Supplementary Fig. 7), which is in good agreement with length change of azobenzene molecule in DFT calculation. Figure 2c shows the typical electrical characteristics of a bare CNT$_B$-CNT$_T$ junction without the molecular layer. Note that metallic CNTs were chosen (Supplementary Fig. 8) to avoid resistance changes in the semiconducting CNTs. The memory measurement was conducted by sweeping V$_{ds}$ = ±10 V from negative to positive values (i-ii) and then back to negative values (iii-iv), indicating no memory window in the $I_{ds}$-$V_{ds}$ curves. In contrast, the CNT$_B$-M/CNT$_T$ vdWI

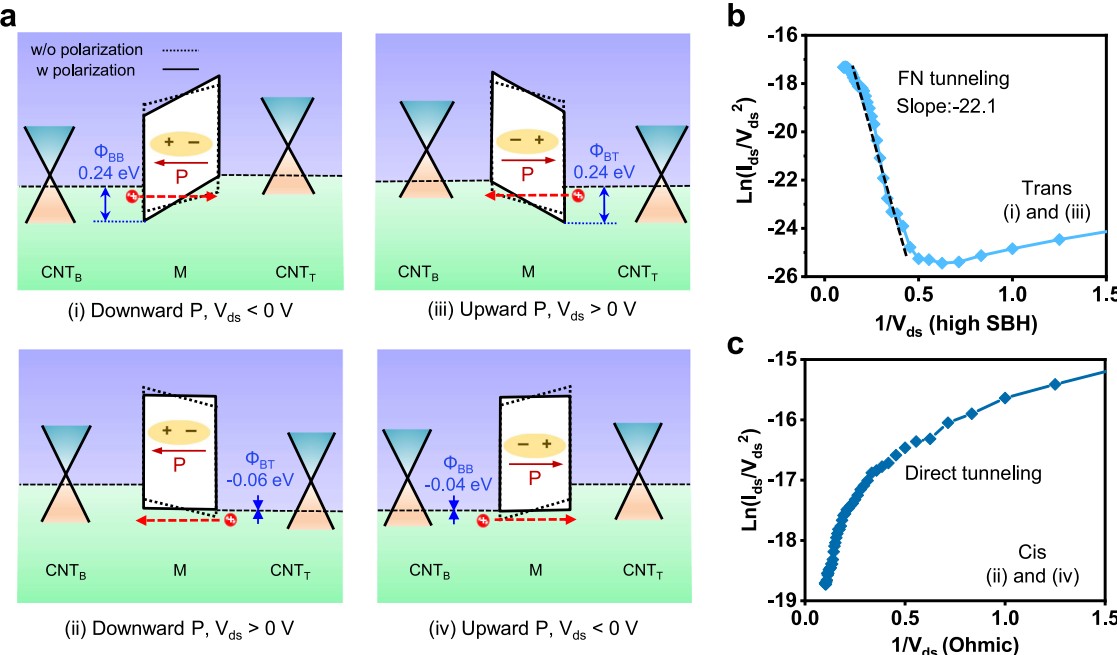

**Fig. 3 | Electrical response and device working mechanism of CNT$_B$-M/CNT$_T$ vdWI. a** Energy band diagrams for the downward polarization (i, ii) and upward polarization (iii, iv) indicating the CNT$_B$-M/CNT$_T$ memory operation. **b, c** Ln($I(V)/V^2$) vs. $1/V$ curve in the (i, iii) and (ii, iv) regimes regarding the *trans* and *cis* states, respectively.

device (Fig. 2d) exhibits a large memory window with a high on/off current ratio of >10$^5$ at V$_{ds}$ = 5 V, which is attributed to the resistive switching between two states, e.g., the *trans*-isomer (assigned as the high-resistance state (HRS)) and *cis*-isomer (assigned as the low-resistance state (LRS))[43]. As shown by our DFT simulation in Fig. 2b, the ferroelectric property of the azobenzene molecule is attributed to the change in direction of the dipole moment of the -CF$_3$ group. In the *trans*-conformation, the dipole is aligned vertically along the axis of the molecule. Vertically applied positive and negative drain biases shift the dipole direction of -CF$_3$ in the azobenzene downward and upward, respectively, resulting in the observed ferroelectric behavior[49]. To prove the ferroelectric behavior in our CNT$_B$-M/CNT$_T$ vdWI, we measured the ferroelectric polarization loops in CNT$_B$-CNT$_T$ and CNT$_B$-M/CNT$_T$ (Fig. 2e) along the drain bias sweeping loop. No polarization loop is observed in the CNT$_B$-CNT$_T$ (red curve), implying no ferroelectric behavior in the bare CNT-CNT junction. On the other hand, the CNT$_B$-M/CNT$_T$ junction shows a clear ferroelectric polarization loop (blue curve). At V$_{ds}$ = −10 V (Fig. 2e (i)), the drain electric field aligns the -CF$_3$ dipole downward, resulting in negative polarization. By sweeping V$_{ds}$ from −10 V to 10 V (Fig. 2e (ii, iii)), the -CF$_3$ dipole shifts upward; therefore, the polarization changes from negative to positive. In contrast, the -CF$_3$ dipole shifts downward by sweeping V$_{ds}$ from 10 V to −10 V (Fig. 2e (iv)), resulting in a polarization change from positive to negative.

## Electrical response and device working mechanism

Figure 3a shows the energy band diagram of the CNT$_B$-M/CNT$_T$ vdWI corresponding to the downward (i and ii) and upward (iii and iv) ferroelectric polarization of azobenzene. It is known that the Fermi level (E$_F$) of graphene[19,25] and m-CNT[50,51] can be shifted by doping or external field due to the finite density of states near the Dirac point, resulting in Schottky barrier height (SBH) change. In our device, the E$_F$ of CNT$_B$ and CNT$_T$ are shifted by the ferroelectric polarization field of the azobenzene molecule[52], resulting in SBH change at CNT$_B$-molecule ($\Phi_{BB}$) and CNT$_T$-molecule ($\Phi_{BT}$) (Supplementary Fig. 9). At the downward polarization (Fig. 3a (i and ii)), the ferroelectric field attracts holes to CNT$_T$ and electrons to CNT$_B$, shifting E$_F$ downward

and upward, respectively. It thus forms a negative $\Phi_{BT}$ (−0.06 eV) and a high $\Phi_{BB}$ (0.24 eV) (Supplementary Fig. 9c and 10). By applying negative V$_{ds}$ (Fig. 3a (i)), hole transport takes place from CNT$_B$ to CNT$_T$, therefore the high $\Phi_{BB}$ at the CNT$_B$-M junction reduces the current flow (Fig. 2d (i)). By applying positive V$_{ds}$ (Fig. 3a (ii)), hole transport occurs freely through the Ohmic $\Phi_{BT}$ at the CNT$_T$-M junction, resulting in high current flow (Fig. 2d (ii)). At the upward polarization (Fig. 3a (iii and iv)), the E$_F$ of CNT$_T$ and CNT$_B$ shifts downward and upward, respectively, resulting in a high $\Phi_{BT}$ (0.24 eV) and a negative $\Phi_{BB}$ (−0.04 eV) (Supplementary Fig. 9d and 10). By applying positive V$_{ds}$ (Fig. 3a (iii)), hole transport is blocked by the high $\Phi_{BT}$ at the CNT$_T$-M junction (Fig. 3a (iii)), resulting in low current flow at Fig. 2d (iii). By applying negative V$_{ds}$ (Fig. 3a (iv)), hole transport is enabled by the Ohmic $\Phi_{BB}$ at the CNT$_B$-M junction (Fig. 3a (iv)), resulting in high current flow at Fig. 2d (iv).

To confirm the mechanism of electron transport through the molecular energy barrier, the $I_{ds}$-$V_{ds}$ curve in Fig. 2d is modeled by the Simmons approximation[53]. The direct tunneling (DT) and Fowler-Nordheim tunneling (FNT) are expressed by the following Eqs. (1, 2)[54]:

$$I_{DT}(V) = \frac{A\sqrt{m\Phi_B}q^2 V_{ds}}{h^2 d} \exp\left[\frac{-4\pi\sqrt{m^*\Phi_B}d}{h}\right] : \text{Direct tunneling} \quad (1)$$

$$I_{FNT}(V) = \frac{Aq^3 m V_{ds}^2}{8\pi h \Phi_B d^2 m^*} \exp\left[\frac{-8\pi\sqrt{2m^*}\Phi_B^{\frac{3}{2}}d}{3hqV_{ds}}\right] : \text{FN tunneling} \quad (2)$$

where $A$, $\Phi_B$, $q$, $m$, $m^*$, $d$, and $h$ are the effective contact area, barrier height, electron charge, free electron mass, effective electron mass, barrier width (molecular length), and Planck's constant, respectively. DT and FNT show logarithmic and negative linear behavior in $ln(I/V^2)$ vs. $1/V$ plots, respectively[55] (Supplementary Fig. 11). At high SBH (Fig. 3a (i) and (iii)), $ln(I/V^2)$ vs. $1/V$ plots shows negative linear curve (Fig. 3b), indicating the FNT of hole carriers through large triangular energy barrier. At the Ohmic SBH (Fig. 3a (ii) and (iv)), $ln(I/V^2)$ vs. $1/V$ plots shows logarithmic curve (Fig. 3c), representing direct tunneling of holes through barrier-free junction (field emission).

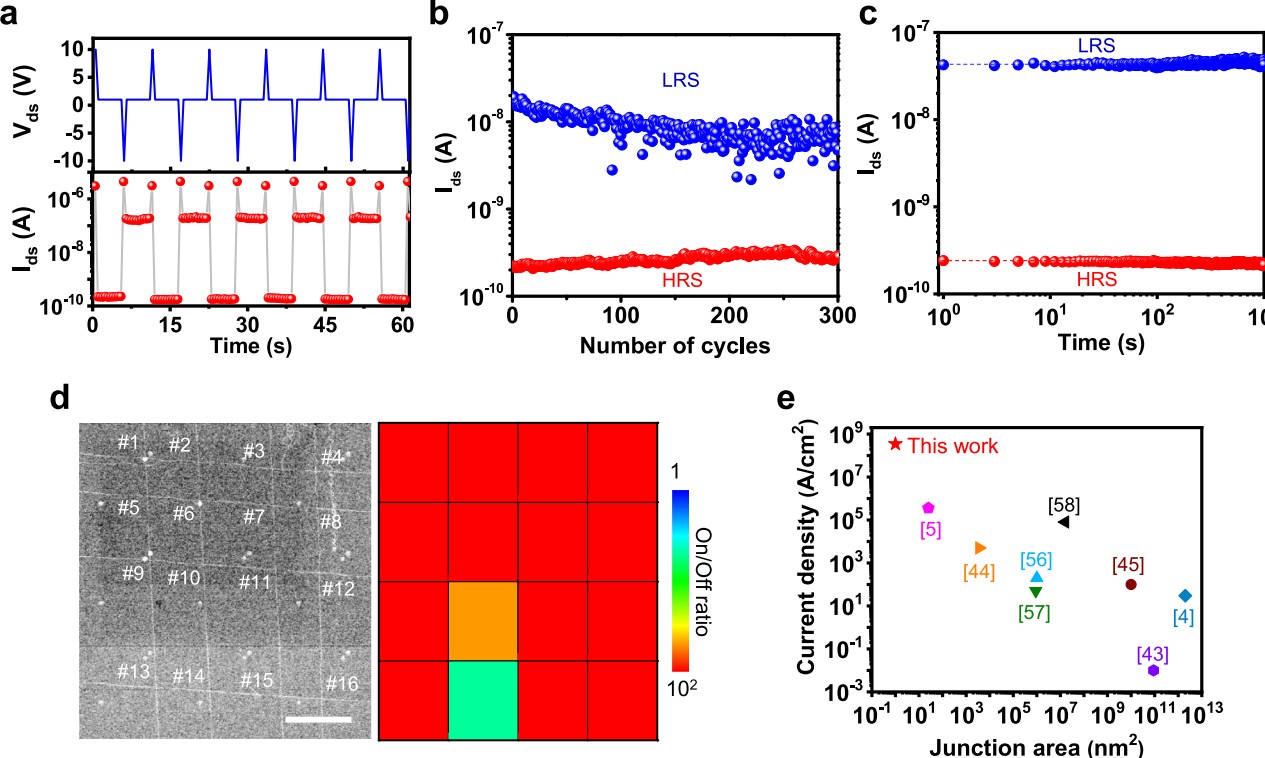

**Fig. 4 | Memory function and microscale integration array of CNT$_B$-M/CNT$_T$ vdWI. a** Electrical response of CNT$_B$-M/CNT$_T$ vdWI memory that repeats writing, reading, erasing, and reading sequence by applying a drain voltage of −10 V, 1 V, + 10 V, and 1 V, respectively. The pulse width was 0.01 s. **b** Endurance and **c** retention characteristic of the CNT$_B$-M/CNT$_T$ device. **d** Cross-staking of 4×4 aligned m-CNTs in CNT$_B$-M/CNT$_T$ vdWI with color mapping of on/off ratios, respectively. Scale bar is 100 μm. **e** Statistical comparison of current density and active area reported for molecule junctions.

## Memory function and microscale integration array

The switching behavior of the azobenzene molecule between the trans and cis states was further confirmed by the electrical response using retention and endurance tests, as shown in Fig. 4a−c. Figure 4a demonstrates a series of memory cycles using the repeated voltage pulses of −10, 1, 10, and 1 V as programing, reading, erasing, and reading operations, respectively. Regular and stable read state was achieved with an ON/OFF ratio over 10³. Our CNT$_B$-M/CNT$_T$ vdWI memory demonstrate an endurance over 300 cycles and a retention >2000 s (Fig. 4b, c), indicating the high reliability of our CNT$_B$-M/CNT$_T$ vdWI memory. Our memory offers a very simple fabrication technique, cross-stacking of aligned CNTs, to achieve microscale array integration of nanoscale vdWI junction memories. Figure 4d shows the integration of 16 vdWI junction memories obtained by cross-stacking 4×4 aligned m-CNTs. Most of vdWI memories exhibited an on/off ratio larger than 100 with a 100% operational yield (Supplementary Fig. 12). Figure 4e shows a comparison of the current density and junction area of the memory. With an ultra-small junction area (1 nm²), our CNT$_B$-M/CNT$_T$ vdWI memory achieves a maximum current density of 3.4 × 10⁸ A/cm², which is approximately 100 times higher than that of previously reported nanoscale molecule junctions (Fig. 4e and Supplementary Table 1)[4,5,42–44,56–58].

## Discussion

In conclusion, we demonstrate a vertically aligned 1D/0D vdWI memory device based on an assembly of azobenzene molecules sandwiched between multi-crossbar m-CNT arrays with ultra-small critical dimensions. The non-volatile memory behavior of the CNT$_B$-M/CNT$_T$ vdWI achieved high on/off current ratio (~10⁵), high stability (300 switching cycles), long retention time (>2000 s), a 100% operational yield in 4 × 4 memory array integration, and ultra-large current density (3.4 × 10⁸ A/cm²) which is 100 times higher than

previously reported. Our platforms can be applied to the array integration of molecular junction memories. Thus, this study can provide guidelines for the assembly of ultra-small devices for future electronics.

## Methods

### Synthesis of m-CNT arrays

Aligned m-CNT arrays were synthesized by catalytic CVD laminar flow using methane as the carbon source. The catalyst solution was prepared using deionized (DI) water as the solvent. In particular, an aqueous solution of iron nitrate (Fe(NO$_3$)$_3$.9H$_2$O, 99.99%, Aldrich), bis (acetyl-acetonato) dioxo-molybdenum (MoO$_2$(acac)$_2$, 99.98%, Aldrich), poly(vinyl pyrrolidone) (Mw 50 000, Aldrich), and alumina nanoparticles were mixed together in DI water, followed by a soni-cation process[59,60]. Next, the catalyst solution was applied to one edge of a 300 nm-thick SiO$_2$/Si substrate, and the substrate was then placed in a horizontal 1-inch (2.54 cm) quartz tube furnace. The catalyst precursor was reduced using a flowing mixture of argon/hydrogen gas at 1000 °C, and then a methane/hydrogen gas was introduced into the furnace to grow aligned m-CNT arrays. Finally, at the end of the growth process, argon gas was applied during cooling to room temperature[61].

### Synthesis of azobenzene diazonium salt

An azobenzene diazonium salt was synthesized according to a previously reported method, as shown in Supplementary Figs. 1–3[62]. First, in a round-bottom flask, 0.3 g of 4-aminoazobenzene (>98%, TCI) was dissolved in tetrahydrofuran (THF, 99.9%) under an inert atmosphere. Second, 2.0 equivalent of boron difluoride diethyl etherate (99.8%, Samchun) (for synthesis grade) was added slowly to the solution, until the color of the solution turned dark red. Third, 1.6 equivalent of iso-pentyl nitrite was added dropwise into the solution. Finally, the

diazonium salt was recrystallized and vacuum-filtered with cold diethyl ether, and the product was dried under vacuum.

## Device fabrication procedure

Aligned $CNT_B$ was transferred from the CVD-grown sample onto a $SiO_2/Si$ substrate (300 nm-thick $SiO_2$) using a wet transfer approach (Supplementary Fig. 13). The poly(methyl methacrylate) (PMMA) solution was spin coated onto $CNT_B$ CVD-grown substrate at 3000 rpm for 60 s (Supplementary Fig. 13a (i-ii)). After baking at 150 °C for 5 min, the sample was then floated onto solution consist of 1% HF and DI water. By etching $SiO_2$ sacrificial layer, $CNT_B$ with the PMMA supporting layer is floated on the etchant solution (Supplementary Fig. 13a (iii)). The $CNT_B$ with PMMA layer was transferred to DI water using Polyethylene terephthalate (PET) holder to clean the HF etchant. By using $SiO_2/Si$ target substrate, pick up the $CNT_B$/PMMA layer and bake at 150 °C for 5 mins to enhance the adhesion between $CNT_B$ and $SiO_2/Si$ wafer (Supplementary Fig. 13a (iv)). Finally, the PMMA supporting layer was removed by dipping sample in acetone for 1 h (Supplementary Fig. 13a (v)).

For the molecule assembly on $CNT_B$ ($CNT_B$-M, Supplementary Fig. 13a (vi)), the $CNT_B$ samples on $SiO_2/Si$ substrates were immersed in a dimethylformamide solution of azobenzene diazonium salt (50 mM) for 48 h in a glove box. The azobenzene monolayer-modified $CNT_B$ was thoroughly washed with dimethylformamide (DMF) solution and then dried overnight under vacuum.

To transfer the $CNT_T$ onto $CNT_B$-M/$SiO_2/Si$ wafer, we used the dry-transfer method as describe in Supplementary Fig. 13b. Firstly, the $CNT_T$ was detached from original substrate as same technique as the $CNT_B$ (Supplementary Fig. 13b (i–iii)). PMMA supporting layer was then flipped-up and floated on DI water using PET holder (Supplementary Fig. 13b (iv–v)). The PMMA layer was then picked up using dry transfer holder with alignment hole (Supplementary Fig. 13b (vi)). The $CNT_T$ was cross aligned and attached above the $CNT_B$ using a X-Y-Z positioner while looking through the hole in the holder under an optical microscope (Supplementary Fig. 13b (vii-viii)). Finally, the $CNT_B$-M/$CNT_T$ crossed structure was obtained by removing PMMA supporting layer using acetone (Supplementary Fig. 13b (ix)).

The source and drain electrodes were patterned by e-beam photolithography (EBL) followed by metal deposition using the e-beam evaporator (EBV) method for Cr/Au (5/50 nm) at $2 \times 10^{-6}$ Torr. Unwanted m-CNTs were etched using a reactive-ion etching (RIE) process under an oxygen gas environment.

## Characterizations

The electrical characteristics were measured using a probe station and source/measure units (Keithley 4200) under high vacuum ($2 \times 10^{-6}$ Torr). Raman spectroscopy was performed using a Witec system at a 532 nm wavelength excitation. Scanning electron microscopy (SEM) (JEOL, JSM-6510) images were taken in the secondary electron image mode at an accelerating voltage of 1 kV. Atomic force microscopy (AFM; SPA-400, SEIKO) was used to record the morphological images using the tapping mode. The FT-IR spectrum was detected using a Bruker VERTEX 70 series spectrometer under the ATR mode in the range of 600–4000 $cm^{-1}$. $^1H$ nuclear magnetic resonance (NMR) spectroscopy was performed using a Bruker Ascend™ 500 system. Photoirradiation with a 360/430 nm laser wavelength was performed using a Power Arc ultraviolet 100 instrument (ultraviolet Process Supp Inc.).

## Density functional theoretical calculations

The structure of the azobenzene molecule was theoretically optimized using density functional calculations under the PBE approximation in the Quantum Espresso code[63]. Two different initial configurations corresponding to cis- and trans- isomers were generated and then optimized to determine the local minimum energy. The energy and

force in the molecule were relaxed until less than $10^{-4}$ Ha and $10^{-3}$ Ha/bohr, respectively. The core potentials of atoms were replaced by ultrasoft and paw potentials from the standard library for solids with a cut-off energy of 400 eV[64,65]. The charge in atoms was calculated by the Löwdin method. A unit cell box of $50 \times 20 \times 20$ Å$^3$ was used to avoid interactions of molecules between cells due to the periodic condition.

## Data availability

All data are available within the Article and Supplementary Files, or available from the corresponding authors on reasonable request. Source data are provided with this paper.

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

## Acknowledgements

This work was supported by the National Research Foundation of Korea (NRF) grant funded by the Korean government (MSIT) (NRF-2021R1A2 C2004027, NRF-2020K1A3A1A05103462, NRF-2020M3A9E4039241, NRF-2021R1A4A1033424, NRF-2018R1A2B6006721 by S.H.S), Multi-Ministry Collaborative R&D Program funded by KNPA, MSIT, MOTIE, ME, and NFA (NRF-2017M3d9A1073539), ICT Creative Consilience program (IITP-2020-0-01821), Samsung Research Funding & Incubation Center of Samsung Electronics (SRFC-MA1701-01), Advanced Facility Center

for Quantum Technology, and the Institute for Basic Science (IBSR011-D1).

## Author contributions

W.J.Y. conceived the research. W.J.Y. and T.L.P. designed the experiments and wrote the manuscript. T.L.P. performed most of the experiments, including synthesizing carbon nanotubes, device fabrication, characterization, and data analysis. S.H.S. and Y.H.C. synthesized and characterized the azobenzene molecule. Q.A.V. assisted in device fabrication. D.L.D. conducted the simulations. Y.H.L. and H.Y.L. participated in the discussion regarding the results. All authors participated in revising the manuscript.

## Competing interests

The authors declare no competing interests
