## [Peer Review File · Nature Communications]

CNT-molecule-CNT (1D-0D-1D) van der Waals integration
ferroelectric memory with 1-nm² junction areaREVIEWER COMMENTS

Reviewer #1 (Remarks to the Author):

The authors present a vertical CNT-molecule-CNT memory with a tiny junction area of 1 nm^2 . Through ferroelectric switching of azobenzene molecules, the device exhibit excellent memory characteristics, including an ultra-long retention performance above 2000 s, an ultra-high on/off ratio of approximately 10^5 , and especially a record high current density. The integrated array devices further confirms the potential for large-scale applications. Considering that the overall quality and significance of the work, the reviewer feels that this paper can be accepted with minor revision:

1. In figure 1e, the erasing-voltage is set to -10 V with a positive reading-voltage. In figure 4a, the erasing-voltage is set to +10 V, however, read voltage is also set to a positive value. Here, the operation mechanisms need to be clarified.
2. It is not necessary to present the 2×2 and 3×3 aligned memories in Figure 4d, which causes an ambiguity about the performance uniformity.
3. In methods, the process for fabricating vertical devices based on CNTs is not adequately described.

Reviewer #2 (Remarks to the Author):

In this manuscript, Phan et. al. have fabricated the vertical 1D-0D-1D vdWI devices composed of cross-stacking top CNTs on a SAM layer over bottom CNTs. The devices exhibit good memory performance due to the ferroelectric switching of the azobenzene molecules, including retention time above 2000 s, large on/off ratio (approximately 105) and high current density ($3.4 \times 10^8 \text{ A/cm}^2$). Array integration of such devices has also been demonstrated.

The experimental results are impressive. However, the quality of the manuscript is not good enough for publication. Most importantly, the abstract does not well correspond with the contents in the main text, and there are two obvious formula errors. These make it difficult to understand the mechanism behind the good performance. More concerns are listed below:

1. In the abstract of this manuscript (line 24-26), the switching of the memory was attributed to the photo-activated cis-trans transformation (ferroelectric switching) of the azobenzene molecules. But the memory in the main text was actually realized by electric field regulation. And, the analysis of the photo-activated cis-trans transformation in Fig.2c and Fig.2d as well as the corresponding text is not closely related to the electric-field-controlled memory. All these descriptions cause confusion and made this paper not suitable for publication, which could only be remedied by a complete re-writing.
2. The authors claimed "the world's smallest junction area of 1 nm^2 ", which could not be true considering a large number of reports on molecular electronics published in the past 30 years, including various CNT-cross-bar structures. Many molecular junctions are of single molecular size, well below 1 nm^2 if one would make a similar claim using the vague junction area definition shown in this manuscript. A very early reference is the book by Jim Tour of Rice Univ: Molecular Electronics, World Scientific, 2003.
3. The three sentences in lines 85-90 contradict the description in lines 76-78. Lines 76-78 say that the molecules self-assembled on the surface of CNTB through π - π interaction, but lines 85-90 say that covalent bonds are formed.
4. Regarding the transport mechanism of the devices, the equation 1 (Line 175) and equation 2 (Line 182) are obviously wrong. Specifically, the exponential factor was left out in equation 1 describing the Direct Tunneling; and multiple V_d s symbols were incorrectly entered in equation 2 describing the FN Tunneling. The authors should correct the equations and check the correctness of the data in Fig.3e and Fig.3f.
5. How do you know the electron is the majority carrier in the CNTB-M/CNTT junctions? At least the LOMO and HOMO of the molecule should be given to make the band diagram in Fig.3d more convincing.
6. The actual morphology of the self-assembled molecule layer on an individual CNT should be

elucidated. In Fig.1 and Fig.2, the molecules are depicted to be in a row on the upper surface of CNT. Obviously, such molecular configuration, oriented perpendicular to the substrate with no lateral support, could not be possible in reality. The detailed information is needed for correctly defining the junction size and explaining transport mechanism.

7. As for the preparation of devices, details of "conventional wet transfer technique" and "well-known dry transfer method" should be explained.

8. If the height of the cross point is different in high and low resistance states (according to the DFT calculation in Fig.2b), it should be measurable by AFM. Additional experiment of such measurement is strongly recommended because it could provide a strong evidence to support the proposed switching mechanism of the device.

Response to Reviewer 1:

General Comments: The authors present a vertical CNT-molecule-CNT memory with a tiny junction area of 1 nm^2 . Through ferroelectric switching of azobenzene molecules, the device exhibits excellent memory characteristics, including an ultra-long retention performance above 2000 s, an ultra-high on/off ratio of approximately 10^5 , and especially a record high current density. The integrated array devices further confirm the potential for large-scale applications. Considering that the overall quality and significance of the work, the reviewer feels that this paper can be accepted with minor revision:

Response: We thank the reviewer for carefully reading our manuscript and valuable comments. We especially appreciate the specific thoughtful questions, and your suggestions are helpful to improve detailed explanations for our manuscript. We would like to address the concern below point-by-point.

1. In figure 1e, the erasing-voltage is set to -10 V with a positive reading-voltage. In figure 4a, the erasing-voltage is set to +10 V, however, read voltage is also set to a positive value. Here, the operation mechanisms need to be clarified.

Response: Thank you for the finding typos. We have corrected the caption for Figure 2e (Figure 4a in revised manuscript), write ($V_{\text{ds}} = -10\text{V}$), erase ($V_{\text{ds}} = +10\text{V}$) and $V_{\text{read}} = 1\text{V}$.

2. It is not necessary to present the 2×2 and 3×3 aligned memories in Figure 4d, which causes an ambiguity about the performance uniformity.

Response: Thank you for this valuable comment. We have removed 2×2 and 3×3 arrays and rearrange the Figure 4 in the revised manuscript.

3. In methods, the process for fabricating vertical devices based on CNTs is not adequately described.

Response: We apologize for the lack of fabrication details. We have updated details of fabrication process in method section "Device fabrication procedure" and Supplementary Figure S11.

Device fabrication procedure. Aligned CNT_B was transferred from the CVD-grown sample onto a SiO_2/Si substrate (300 nm-thick SiO_2) using a wet transfer approach (Figure S11). The poly(methyl methacrylate) (PMMA) solution was spin coated onto CNT_B CVD-grown substrate at 3000 rpm for 60 s (Figure S11a (i-ii)). After baking at 150°C for 5 min, the sample was then floated onto solution consist of 1% HF and DI water. By etching SiO_2 sacrificial layer, CNT_B with the PMMA supporting layer is floated on the etchant solution (Figure S11a (iii)). The CNT_B with PMMA layer was transferred to DI water using Polyethylene terephthalate (PET) holder to clean the HF etchant. By using SiO_2/Si target substrate, pick up the CNT_B/PMMA layer and bake at 150°C for 5 mins to enhance the adhesion between CNT_B and SiO_2/Si wafer (Figure S11a (iv)). Finally, the PMMA supporting layer was removed by dipping sample in acetone for 1 hour (Figure S11a (v)).

For the molecule assembly on CNT_B ($\text{CNT}_B\text{-M}$, Figure S11a (vi)), the CNT_B samples on SiO_2/Si substrates were immersed in a dimethylformamide solution of azobenzene diazonium salt (50 mM) for 48

h in a glove box. The azobenzene monolayer-modified CNT_B was thoroughly washed with dimethylformamide (DMF) solution and then dried overnight under vacuum.

To transfer the CNT_T onto CNT_B -M/ SiO_2 /Si wafer, we used the dry-transfer method as describe in Figure S11b. Firstly, the CNT_T was detached from original substrate as same technique as the CNT_B (Figure S11b(i-iii)). PMMA supporting layer was then flipped-up and floated on DI water using PET holder (Figure S11b (iv-v)). The PMMA layer was then picked up using dry transfer holder with alignment hole (Figure S11b (vi)). The CNT_T was cross aligned and attached above the CNT_B -M/ SiO_2 /Si wafer using a X-Y-Z positioner while looking through the hole in the holder under an optical microscope (Figure S11b (vii-viii)). Final the CNT_B -M/ CNT_T crossed structure was obtained by removing PMMA supporting layer using acetone (Figure S11b (ix)).

The source and drain electrodes were patterned by e-beam photolithography (EBL) followed by metal deposition using the e-beam evaporator (EBV) method for Cr/Au (5/50 nm) at 2×10^{-6} Torr. Unwanted m-CNTs were etched using a reactive-ion etching (RIE) process under an oxygen gas environment.

Supplementary Figure S11 CNT_B-M/CNT_T vdWI fabrication process. a Wet-transfer technique used for CNT_B. **b** Dry transfer technique to align CNT_T on CNT_B-M to form the CNT_B-M/CNT_T vdWI device.

Overall, we appreciate the thoughtful comments and valuable suggestions from the reviewer. We have made necessary revisions to fully address the concerns raised by the reviewers, which greatly improved the manuscript. We believe that our study represents an important advancement in this field and should make a valuable contribution to Nature Communications.

Response to Reviewer 2:

General Comments: In this manuscript, Phan et. al. have fabricated the vertical 1D-0D-1D vdWI devices composed of cross-stacking top CNTs on a SAM layer over bottom CNTs. The devices exhibit good memory performance due to the ferroelectric switching of the azobenzene molecules, including retention time above 2000 s, large on/off ratio (approximately 10^5) and high current density (3.4×10^8 A/cm²). Array integration of such devices has also been demonstrated. The experimental results are impressive. However, the quality of the manuscript is not good enough for publication. Most importantly, the abstract does not well correspond with the contents in the main text, and there are two obvious formula errors. These make it difficult to understand the mechanism behind the good performance. More concerns are listed below:

Response: We thank the reviewer for carefully reading our manuscript and valuable comments. We especially appreciate the reviewer described our experimental results as “impressive”, and specific thoughtful questions. Reviewer’s suggestions are very helpful to improve detailed explanations for our manuscript. We would like to address the concern below point-by-point.

1. In the abstract of this manuscript (line 24-26), the switching of the memory was attributed to the photo-activated cis-trans transformation (ferroelectric switching) of the azobenzene molecules. But the memory in the main text was actually realized by electric field regulation. And, the analysis of the photo-activated cis-trans transformation in Fig.2c and Fig.2d as well as the corresponding text is not closely related to the electric-field-controlled memory. All these descriptions cause confusion and made this paper not suitable for publication, which could only be remedied by a complete re-writing.

Response: Thank you for this valuable comment. To avoid the confusion, we have removed the photo-activated cis-trans transformation. Instead, we included the reference paper (Ref. 48, PRL 96, 156106, 2006) that first introduced the cis-trans transition by electric field. It is revised in line110-112. “It is known that the azobenzene molecules transformed between *cis* and *trans* states by external bias⁴⁸, inducing the ferroelectric switching memory behavior in CNT_B-M/CNT_T vdWI.”

2. The authors claimed “the world's smallest junction area of 1 nm²”, which could not be true considering a large number of reports on molecular electronics published in the past 30 years, including various CNT-cross-bar structures. Many molecular junctions are of single molecular size, well below 1 nm² if one would make a similar claim using the vague junction area definition shown in this manuscript. A very early reference is the book by Jim Tour of Rice Univ: Molecular Electronics, World Scientific, 2003.

Response: Thank you for valuable comment. We have changed the word “the world’s smallest junction area of 1 nm²” to “ultra-small junction area of 1 nm²”. We also included the reference book at Ref. 6.

3. The three sentences in lines 85-90 contradict the description in lines 76-78. Lines 76-78 say that the molecules self-assembled on the surface of CNT_B through π - π interaction, but lines 85-90 say that covalent bonds are formed.

Response: We apologize for the mistakes and thank the reviewer for careful reading and finding such mistakes. It is covalent C-C sp^3 bonds between carbon atoms C=C sp^2 of CNT_B and aryl radicals (azobenzene molecular). We have corrected the line 76-78 to “The CNT_B is covalently functionalized by the azobenzene molecules that allow ultra-thin mono-molecular self-assembly⁴³ (Fig. 1a (ii), details are provided in the Methods section and Supplementary Figs. S1–S3).”

4. Regarding the transport mechanism of the devices, the equation 1 (Line 175) and equation 2 (Line 182) are obviously wrong. Specifically, the exponential factor was left out in equation 1 describing the Direct Tunneling; and multiple V_d symbols were incorrectly entered in equation 2 describing the FN Tunneling. The authors should correct the equations and check the correctness of the data in Fig.3e and Fig.3f.

Response: We apologize for the mistakes and thank the reviewer for careful reading and finding such mistakes. We have corrected the equations at 8th page in revised manuscript. We only made mistake to writing equations, but the graph was correctly fitted using correct equation ($\ln(I(V)/V^2)$ vs. $1/V$ plots). Therefore, tunneling mechanism in our CNT-M/CNT vdWI is valid.

5. How do you know the electron is the majority carrier in the CNT_B-M/CNT_B junctions? At least the LOMO and HOMO of the molecule should be given to make the band diagram in Fig.3d more convincing.

Response: Thank you for this valuable comment. We have calculated the HOMO and LUMO levels of Trans (HOMO: -5.1 eV, LUMO: -3.2 eV) and Cis state (HOMO: -5.0 eV, LUMO: -3.1 eV) by DFT calculation. Based on these HOMO, LUMO levels and work function of SWCNT (4.5 eV, Ref 50). Because the barrier height between m-CNT and LUMO (0.5~0.6 eV) is lower than m-CNT and HOMO (1.6~1.7 eV), majority carriers are hole carriers. We have changed the schematic band diagrams (Fig. 3a) and corresponding texts (lines 136-141).

6. The actual morphology of the self-assembled molecule layer on an individual CNT should be elucidated. In Fig.1 and Fig.2, the molecules are depicted to be in a row on the upper surface of CNT. Obviously, such molecular configuration, oriented perpendicular to the substrate with no lateral support, could not be possible in reality. The detailed information is needed for correctly defining the junction size and explaining transport mechanism.

Response: Thank you for this valuable comment. We agree that the aligned molecules in a single row are misleading. In previous schematic images, for easy understanding of CNT-molecule-CNT structure, we draw only a single adsorbate in the direction perpendicular to the nanotube long axis. In a realistic CNT-molecule-CNT structure, however, the molecule must bond not only to top wall of CNT_B, but also to side wall of CNT_B, which is structure of molecules surrounding the CNT [Ref 47]. The height change of CNT_B is about 2.5~2.9 nm, which is in good agreement with the height of azobenzene molecule in DFT calculation (2.8 nm). Therefore, we can expect the top molecule stand vertically by support of side molecules. We have changed the schematic image of CNT-molecule-CNT structure Figure 1a-(iii) to molecules surrounding the CNT_B, and revised corresponding texts (Lines 97-102).

7. As for the preparation of devices, details of "conventional wet transfer technique" and "well-known dry transfer method" should be explained.

Response: We apologize for the lack of fabrication details. We have updated details of fabrication process in method section "Device fabrication procedure" and Supplementary Figure S11.

Device fabrication procedure. Aligned CNT_B was transferred from the CVD-grown sample onto a SiO₂/Si substrate (300 nm-thick SiO₂) using a wet transfer approach (Figure S11). The poly(methyl methacrylate) (PMMA) solution was spin coated onto CNT_B CVD-grown substrate at 3000 rpm for 60 s (Figure S11a (i-ii)). After baking at 150 °C for 5 min, the sample was then floated onto solution consist of 1% HF and DI water. By etching SiO₂ sacrificial layer, CNT_B with the PMMA supporting layer is floated on the etchant solution (Figure S11a (iii)). The CNT_B with PMMA layer was transferred to DI water using Polyethylene terephthalate (PET) holder to clean the HF etchant. By using SiO₂/Si target substrate, pick up the CNT_B/PMMA layer and bake at 150 °C for 5 mins to enhance the adhesion between CNT_B and SiO₂/Si wafer (Figure S11a (iv)). Finally, the PMMA supporting layer was removed by dipping sample in acetone for 1 hour (Figure S11a (v)).

For the molecule assembly on CNT_B (CNT_B-M, Figure S11a (vi)), the CNT_B samples on SiO₂/Si substrates were immersed in a dimethylformamide solution of azobenzene diazonium salt (50 mM) for 48 h in a glove box. The azobenzene monolayer-modified CNT_B was thoroughly washed with dimethylformamide (DMF) solution and then dried overnight under vacuum.

To transfer the CNT_T onto CNT_B-M/SiO₂/Si wafer, we used the dry-transfer method as describe in Figure S11b. Firstly, the CNT_T was detached from original substrate as same technique as the CNT_B (Figure S11b(i-iii)). PMMA supporting layer was then flipped-up and floated on DI water using PET holder (Figure S11b (iv-v)). The PMMA layer was then picked-up using dry transfer holder with alignment hole (Figure S11b (vi)). The CNT_T was cross aligned and attached above the CNT_B-M/SiO₂/Si wafer using a X-Y-Z positioner while looking through the hole in the holder under an optical microscope (Figure S11b (vii-viii)). Final the CNT_B-M/CNT_T crossed structure was obtained by removing PMMA supporting layer using acetone (Figure S11b (ix)).

The source and drain electrodes were patterned by e-beam photolithography (EBL) followed by metal deposition using the e-beam evaporator (EBV) method for Cr/Au (5/50 nm) at 2×10^{-6} Torr. Unwanted m-CNTs were etched using a reactive-ion etching (RIE) process under an oxygen gas environment.

Supplementary Figure. S11 CNT_B-M/CNT_T vdWI fabrication process. a Wet-transfer technique used for CNT_B. **b** Dry transfer technique to align CNT_T on CNT_B-M to form the CNT_B-M/CNT_T vdWI device.

8. If the height of the cross point is different in high and low resistance states (according to the DFT calculation in Fig.2b), it should be measurable by AFM. Additional experiment of such measurement is strongly recommended because it could provide a strong evidence to support the proposed switching mechanism of the device.

Response: Thank you for this valuable comment. The electric field-induced trans-cis transition of azobenzene was already introduced in a previous journal (Ref. 48: PRL 96, 156106, 2006). We have measured the thickness change of trans (6.31 ~ 6.45 nm in Figure S7a-c) and cis states (5.65~5.86 nm in Figure S7d-f). As a result, our CNT_B-M/CNT_T vdWI memory showed a ~0.6 nm thickness change, which is in good agreement with length change of azobenzene molecule in DFT calculation. In revised manuscript, we have referred the electric field-induced trans-cis transition of azobenzene (Ref. 48) in line 110-114. **“It is known that the azobenzene molecules transformed between *cis* and *trans* states by external bias⁴⁸, inducing the ferroelectric switching memory behavior in CNT_B-M/CNT_T vdWI. The thickness change of CNT_B-M/CNT_T vdWI of trans-cis transitions of azobenzene was measured to ~0.6 nm (Fig. S7), which is in good agreement with length change of azobenzene molecule in DFT calculation.”**

Supplementary Figure. S7 Thickness change of CNT_B-M/CNT_T vdWI by trans-cis transition of azobenzene. AFM image and height profile distribution of CNT_B-M/CNT_T vdWI at (a-c) HRS, and (d-f) LRS, respectively. The line profiles are displayed as cross-bar line as in (a, d).

Overall, we appreciate the thoughtful comments and valuable suggestions from the reviewer. We have made necessary revisions to fully address the concerns raised by the reviewers, which greatly improved the manuscript. We believe that our study represents an important advancement in this field and should make a valuable contribution to Nature Communications.

REVIEWER COMMENTS

Reviewer #1 (Remarks to the Author):

After the authors answer the reviewer's questions and revise the manuscript point to point, the technical presentation is good. The reviewer considers the authors have revised the manuscript to a level of publication. Therefore the reviewer recommends the manuscript be accepted.

Reviewer #2 (Remarks to the Author):

Although the authors have made many corrections in the revised manuscript, the key part on transport mechanism analysis for the devices has not been satisfactorily revised, still with many errors and vague analysis. As a revision, the authors calculated the LOMO and HOMO of Trans and Cis states of the molecule and confirmed that the majority carrier is hole instead of electron as described in the original manuscript. This is a huge change. However, the band diagram in Fig. 3a, the transport mechanism has not been modified to a satisfactory level for publication.

Specifically, firstly, the position of the CNT's Fermi level on each side should be fixed relative to the HOMO edge for each specific polarization of the molecules, regardless of the applied bias voltage. But this is not the case in Fig. 3a. A proper band diagram should also show a smaller hole barrier in Fig. 3a (ii) and (iv) than those in Fig. 3a (i) and (iii). However, it is still questionable whether the cases corresponding to Fig. 3a (i) and (iii) or Fig. 3a (ii) and (iv) should follow Fowler-Nordheim tunneling (FNT) mechanism or direct tunneling (DT) mechanism, or even thermal emission mechanism. They should be determined by the detailed positions of the CNT's Fermi levels relative to the HOMO edges. Secondly, according to equations 1 and 2 in Lines 165-166, $\ln(I/V^2)-1/V$ plots should be used to fit the data corresponding to FNT mechanism and $\ln(I/V)-1/V$ plots should be used for DT mechanism. However, as stated in the caption, the $\ln(I/V^2)-1/V$ plots are used in both Fig. 3b and Fig. 3c. Considering the obvious errors in the two basic equations in the original manuscript and the error in the revised manuscript, I do not have the confidence in the correctness of data analysis in this manuscript.

Finally, this paper discussed the CNTB-M/CNTT heterostructure. However, in Line 170 of the manuscript, it says "we estimated the barrier heights between graphene/h-BN and h-BN/MoS2 using the FNT equation". Why graphene/h-BN and h-BN/MoS2 that are irrelevant to this manuscript? It is difficult to imagine how such error could occur if the manuscript were written by serious scientists.

Response to Reviewer 2:

General Comments: *Although the authors have made many corrections in the revised manuscript, the key part on transport mechanism analysis for the devices has not been satisfactorily revised, still with many errors and vague analysis. As a revision, the authors calculated the LOMO and HOMO of Trans and Cis states of the molecule and confirmed that the majority carrier is hole instead of electron as described in the original manuscript. This is a huge change. However, the band diagram in Fig. 3a, the transport mechanism has not been modified to a satisfactory level for publication.*

Response: We thank the reviewer for carefully reading our manuscript and valuable comments. We especially appreciate the specific thoughtful questions, and your suggestions are helpful to improve detailed explanations for our manuscript. We would like to address the concern below point-by-point.

Q1) Specifically, firstly, the position of the CNT's Fermi level on each side should be fixed relative to the HOMO edge for each specific polarization of the molecules, regardless of the applied bias voltage. But this is not the case in Fig. 3a. A proper band diagram should also show a smaller hole barrier in Fig. 3a (ii) and (iv) than those in Fig. 3a (i) and (iii). However, it is still questionable whether the cases corresponding to Fig. 3a (i) and (iii) or Fig. 3a (ii) and (iv) should follow Fowler-Nordheim tunneling (FNT) mechanism or direct tunneling (DT) mechanism, or even thermal emission mechanism. They should be determined by the detailed positions of the CNT's Fermi levels relative to the HOMO edges.

Response: Thank you for the valuable comment. For more clarity of barrier height, we added the schematics of energy barriers without drain bias in revised Supplementary Fig. S9 with detail explains of energy barrier forming, and also calculated the energy barrier height at CNT_B-molecule and CNT_T-molecule in revised Supplementary Fig. S10. Corresponding text in main manuscript is also revised (139~158).

Supplementary Fig. S9a shows hole energy barrier at CNT_B-molecule and CNT_T-molecule in vacuum, ignoring ferroelectric polarization. Top- and bottom-hole barriers are 0.6 eV, calculated as the difference between HOMO level of molecule (5.1 eV) and Fermi level (E_F) of CNT (4.5 eV). Supplementary Fig. S9b shows the barrier change by oxygen doping in air condition. **It is known that the E_F of graphene [Nat. Mater. 12, 246–252 (2013), Nat. Nanotechnol. 8, 952–958 (2013)] and m-CNT [Proc. Natl. Acad. Sci. 119, e2119016119 (2022), Adv. Mater. 29, 1–8 (2017)] can be shifted by doping or external field due to the finite density of states near the Dirac point, resulting in Schottky barrier height (Φ_B) change.** In our device, oxygen in the air donates hole carriers to the m-CNT [Nanotechnology 16, 1048–1052 (2005)], shifting E_F downwards (4.95~5.05 eV) [Carbon 39, 1913–1917 (2001)] and reducing the barrier height.

Supplementary Fig. S9b and 9c show the barrier change under downward and upward ferroelectric polarization of azobenzene molecule, respectively. As same reason as above, the ferroelectric dipole of azobenzene molecule shifts the E_F of CNT_B and CNT_T due to finite density of states near the Dirac point, resulting in Φ_B change. In our CNT_B-M/CNT_T device, ferroelectric dipole of azobenzene molecule shifts the E_F of CNT_B and CNT_T and changes Schottky barrier height. At the downward polarization (trans state, Supplementary Fig. S9c), electrons and holes

are attracted to CNT_B and CNT_T by polarization field, resulting in upshifts and downshifts of E_F in CNT_B and CNT_T, respectively. Corresponding Schottky barrier height of CNT_B-Molecule (Φ_{BB}) and CNT_T-molecule (Φ_{BT}) increases and decreases by the E_F shift (Φ_B = 0.1 eV + ΔE_F), respectively. The Φ_{BB} and Φ_{BT} are measured to 0.24 eV and -0.06 eV, respectively (Supplementary Fig. S10). At the upward polarization (cis state, Supplementary Fig. S9d), Φ_{BB} and Φ_{BT} are decreased to -0.04 eV and increased to 0.24 eV, respectively, by the same mechanism (Supplementary Fig. S10).

Supplementary Fig. S9 | **a-b** Energy band diagrams of CNT_B-M/CNT_T device without polarization in (a) vacuum and (b) air. Φ_{BB} and Φ_{BT} are Schottky barrier height at CNT_B-molecule and CNT_T-molecule junctions, respectively. **c-d** Energy band diagram of CNT_B-M/CNT_T device under (c) downward (trans) and (d) upward (cis) polarization.

For more accuracy of energy band diagram, we have calculated the Φ_{BB} and Φ_{BT} from current-voltage curve (Supplementary Fig. S10) [Book: Semiconductor material and device characterization. Hoboken, New Jersey, USA: John Wiley & Sons, Inc. (2006)]. The thermionic current-voltage relationship of a Schottky barrier, neglecting series and shunt resistance, is given by

$$I = AA^*T^2 \exp\left[\frac{-q\Phi_B}{kT}\right] \left(\exp\left[\frac{qV}{nkT}\right] - 1 \right) = I_s \left(\exp\left[\frac{qV}{nkT}\right] - 1 \right)$$

Where I_s is the saturation current (obtained at Fig. S10), A the diode area (1 nm²), A^* richardson's constant (120 A/cm²K²), Φ_B the effective barrier height, q the electron charge, n the ideality factor, T the temperature, and V the applied drain voltage.

From $I_s = AA^*T^2 \exp\left[\frac{-q\Phi_B}{kT}\right]$, the Schottky barrier height is calculated to

$$\Phi_B = \frac{kT}{q} \ln \left(\frac{AA^*T^2}{I_s} \right)$$

At the reverse bias ($V_{ds} < 0V$), Φ_{BB} is 0.24 eV (i. downward P) and -0.04 eV (iv. upward P). At the forward bias ($V_{ds} > 0V$), Φ_{BT} is -0.06 eV (ii. downward P) and 0.24 eV (iii. upward P).

Supplementary Fig. S10 Saturation currents (I_s) for Schottky barrier calculation at (a) reverse and (b) forward region. Dashed lines are expansions of the linear region in the I - V curve, which intercept at $V_{ds} = 0$ V is saturation current density (I_s).

At high SBH (0.24 eV in Fig. 3a (i) and (iii)), $\ln(I/V^2)$ vs. $1/V$ plots shows negative linear curve (Fig. 3b), indicating the FNT of hole carriers through large triangular energy barrier. At the Ohmic SBH (-0.04~0.06 eV in Fig. 3a (ii) and (iv)), $\ln(I/V^2)$ vs. $1/V$ plots shows logarithmic curve (Fig. 3c), representing direct tunneling of holes through barrier-free junction (field emission). This is revised in main manuscript, lines 167-171.

Fig. 3 Electrical response and device working mechanism of CNT_B-M/CNT_T vdWI. **a** Energy band diagrams for the downward polarization (i, ii) and upward polarization (iii, iv) indicating the CNT_B-M/CNT_T memory operation. **b-c** $\ln(I(V)/V^2) - 1/V$ curve in the (i, iii) and (ii, iv) regimes regarding the *trans* and *cis* states, respectively.

Q2) Secondly, according to equations 1 and 2 in Lines 165-166, $\ln(I/V^2)$ - $1/V$ plots should be used to fit the data corresponding to FNT mechanism and $\ln(I/V)$ - $1/V$ plots should be used for DT mechanism. However, as stated in the caption, the $\ln(I/V^2)$ - $1/V$ plots are used in both Fig. 3b and Fig. 3c. Considering the obvious errors in the two basic equations in the original manuscript and the error in the revised manuscript, I do not have the confidence in the correctness of data analysis in this manuscript.

Response: We sorry for the insufficient explanation. The fitting of direct tunneling and FN tunneling is introduced by Simmon *et al.* (*J. Appl. Phys.* 34, 1793 (1963)) and more easily explained by Ikuno *et al.* (*APPLIED PHYSICS LETTERS* 99, 023107 (2011)). According to reference papers, direct tunneling and FN tunneling show logarithmic and negative linear behavior in $\ln(I/V^2)$ vs. $1/V$ plots. We have revised it in Supplementary Fig. S11 and corresponding text in main manuscript (159-167)

“The direct tunneling (DT) and Fowler-Nordheim tunneling (FNT) are expressed by the following equations:

$$I_{DT}(V) = \frac{A\sqrt{m\phi_B}q^2V_{ds}}{h^2d} \exp\left[\frac{-4\pi\sqrt{m^*\phi_B}d}{h}\right] : \text{Direct tunneling}$$

$$I_{FNT}(V) = \frac{Aq^3mV_{ds}^2}{8\pi h\phi_B d^2 m^*} \exp\left[\frac{-8\pi\sqrt{2m^*}\phi_B^{\frac{3}{2}}d}{3hqV_{ds}}\right] : \text{FN tunneling}$$

where A , ϕ_B , q , m , m^* , d , and h are the effective contact area, barrier height, electron charge, free electron mass, effective electron mass, barrier width (molecular length), and Planck’s constant, respectively.

The relation between tunneling current (I) and the applied voltage (V_{ds}) can be expressed to [*APPLIED PHYSICS LETTERS* 99, 023107 (2011)]

$$I \propto \begin{cases} V_{ds} \exp\left[\frac{-4\pi\sqrt{m^*\phi_B}d}{h}\right] : \text{Direct tunneling} \\ V_{ds}^2 \exp\left[\frac{-8\pi\sqrt{2m^*}\phi_B^{\frac{3}{2}}d}{3hqV_{ds}}\right] : \text{FN tunneling} \end{cases}$$

By dividing V_{ds}^2 and making $\ln()$ in the equation, the relation between I and V_{ds} become

$$\ln\left(\frac{I}{V^2}\right) \propto \begin{cases} \ln\left(\frac{1}{V}\right) : \text{Direct tunneling} \\ -\left(\frac{1}{V}\right) : \text{FN tunneling} \end{cases}$$

Therefore, direct tunneling and FN tunneling show logarithmic and negative linear behavior in $\ln(I/V^2)$ vs. I/V plots (Fig. 3b and c).”

Q3) Finally, this paper discussed the CNTB-M/CNTT heterostructure. However, in Line 170 of the manuscript, it says "we estimated the barrier heights between graphene/h-BN and h-BN/MoS2 using the FNT equation". Why graphene/h-BN and h-BN/MoS2 that are irrelevant to this manuscript? It is difficult to imagine how such error could occur if the manuscript were written by serious scientists.

Response: We accidentally submitted a non-final version. We apologize for unprofessional such mistakes. We have deleted these sentences in revised main manuscript, lines 159-167.

“To confirm the mechanism of electron transport through the molecular energy barrier, the I_{ds} - V_{ds} curve in Fig. 2d is modeled by the Simmons approximation⁵³. The direct tunneling (DT) and Fowler-Nordheim tunneling (FNT) are expressed by the following equations⁵⁴:

$$I_{DT}(V) = \frac{A\sqrt{m\Phi_B}q^2V_{ds}}{h^2d} \exp\left[\frac{-4\pi\sqrt{m^*}\Phi_B d}{h}\right] : \text{Direct tunneling}$$
$$I_{FNT}(V) = \frac{Aq^3mV_{ds}^2}{8\pi h\Phi_B d^2 m^*} \exp\left[\frac{-8\pi\sqrt{2m^*}\Phi_B^{\frac{3}{2}}d}{3hqV_{ds}}\right] : \text{FN tunneling}$$

where A , Φ_B , q , m , m^* , d , and h are the effective contact area, barrier height, electron charge, free electron mass, effective electron mass, barrier width (molecular length), and Planck’s constant, respectively. DT and FNT show logarithmic and negative linear behavior in $\ln(I/V^2)$ vs. I/V plots, respectively⁵⁵ (Supplementary Fig. S11).”

Overall, we appreciate the thoughtful comments and valuable suggestions from the reviewer. We have made necessary revisions to fully address the concerns raised by the reviewers, which greatly improved the manuscript. We believe that our study represents an important advancement in this field and should make a valuable contribution to Nature Communications.

REVIEWERS' COMMENTS

Reviewer #2 (Remarks to the Author):

I have read this revised version of manuscript with a great relief that all my previous concerns have been well addressed. It's now ready to go.